# AUTOMATIC PARAMETER TYING IN NEURAL NETWORKS

## ABSTRACT

Recently, there has been growing interest in methods that perform neural network compression, namely techniques that attempt to substantially reduce the size of a neural network without performance degradation. In this paper, we propose a simple compression algorithm that incorporates pruning within scalar quantization, whose non-probabilistic nature results in minimal change and overhead to standard network training. The key idea in our approach is to modify the original optimization problem by adding $K$ independent Gaussian priors and a sparsity-inducing prior over the parameters. We show that our approach is easy to implement using existing neural network libraries, generalizes $\ell_1$ and $\ell_2$ regularization, and elegantly enforces parameter tying/pruning constraints on all parameters across the network. Experimentally, we demonstrate that our method yields state-of-the-art compression on several standard benchmarks with minimal loss in accuracy while requiring significantly less hyperparameter tuning compared with related, competing approaches.

## 1 INTRODUCTION

Neural networks represent a family of highly flexible and scalable models that have rapidly achieved state-of-the-art performance in diverse domains including computer vision (Krizhevsky et al., 2012; Girshick et al., 2014; He et al., 2016), speech (Hinton et al., 2012; Deng et al., 2013), and sentiment analysis (Glorot et al., 2011). Despite their successes, the storage requirements of large, modern neural networks make them impractical for certain applications with storage limitations (e.g., mobile devices). Moreover, as they are often trained on small datasets compared to their number of parameters (typically in the millions for state-of-the-art models), they can potentially overfit. In recent work, Denil et al. (2013) showed that a large proportion of neural network parameters are in fact not required for their generalization performance, and interest in model compression has surged.

A variety of methods have been proposed to perform compression including pruning (LeCun et al., 1990; Han et al., 2015), quantization (Han et al., 2016; K. Ullrich, 2017; Chen et al., 2015), low-rank approximation (Denil et al., 2013; Denton et al., 2014; Jaderberg et al., 2014), group lasso (Wen et al., 2016), variational dropout (Molchanov et al., 2017), teacher-student training (Romero et al., 2014), etc. Here, we focus on the quantization/parameter tying approach to compression combined with pruning. Parameter tying assumptions occur naturally in the construction of convolutional neural networks (CNNs), but in these applications, the parameters to be tied are usually selected in advance of training. Recent work has focused on automatic parameter tying, i.e., automatically discovering which parameters of the model should be tied together. Nowlan & Hinton (1992) proposed a soft parameter tying scheme based on a mixtures of Gaussians prior and suggested a gradient descent method to jointly optimize both the weights in the network and the parameters of the mixture model. Chen et al. (2015) proposed a random parameter tying scheme based on hashing functions. Han et al. (2016) proposed a compression pipeline that involved thresholding to prune low-magnitude parameters, $k$-means clustering to tie parameters layer-wise, and a final retraining stage to fine-tune tied parameters. This work demonstrated that high compression rates are achievable without much loss in accuracy. Building on the work of (Nowlan & Hinton, 1992), K. Ullrich (2017) imposed a Gaussian mixture prior on the parameters to encourage clustering. At convergence, they proposed clustering the weights by assigning them to the mixture component that generates each weight with highest probability. Louizos et al. (2017) proposed a full Bayesian

approach to compression using scale mixture priors. This approach has the advantage that posterior distributions can be used to estimate the significance of individual bits in the learned weights. Louizos et al. (2017) demonstrated that this approach can yield state-of-the-art compression results for some problems. Agustsson et al. (2017) recently proposed a soft-to-hard quantization approach in which scalar quantization is gradually learned through annealing a softened version of quantization distortion; compression is achieved with low-entropy parameter distribution instead of pruning.

While much of the previous work has demonstrated that significant compression can be achieved while preserving the accuracy of the final network (in many cases $\approx 1\%$ loss in accuracy), many of these approaches have potential drawbacks that can limit their applications. The Gaussian mixture approach of Nowlan & Hinton (1992) and K. Ullrich (2017) can be computationally expensive, as the time and memory requirements for backpropagation is increased $K$-fold under a $K$-component GMM prior, in addition to its large number of sensitive hyperparameters that can require extensive tuning. Moreover, the GMM objective itself suffers from well known local (and often pathological) minima issues. These local minimas are in addition to the ones encountered while training a neural network which in turn incurs high computational cost. The approach of Han et al. (2016) uses separate pruning and parameter tying stages, which potentially limits its compression efficiency; additionally, the required layer-wise codebook storage can become expensive especially for deep networks. The parameter tying approach of Chen et al. (2015) is also only applied layerwise, and it typically requires more clusters, i.e., larger $K$, before the random weight sharing is effective (our experiments confirm that random parameter tying yields poor results when the number of distinct parameters is too small). The soft-to-hard quantization approach of (Agustsson et al., 2017) resembles our method, but is essentially probabilistic like the GMM prior as it uses soft assignment for quantization which can be expensive. Finally, the full Bayesian approach, similar to the GMM approach, has a number of additional parameters to tune (e.g., constraints on variances, careful initialization of each of the variational parameters, etc.). The Bayesian approach also requires sampling for prediction (which can be done deterministically but with some additional loss). We hope to argue that such sophisticated methods may not be necessary to achieve good compression in practice.

The approach to compression in this work uses quantization and sparsity inducing priors. For quantization, we consider an independent Gaussian prior, that is, each parameter is non-probabilistically assigned to one of $K$ independent Gaussian distributions, and the prior penalizes each weight by its $\ell_2$ distance to the mean of its respective Gaussian. This prior places no restriction on which weights can be tied together (e.g., weights from the input could be tied to weights into the output), reduces the number of hyperparameters that need to be tuned compared to probabilistic methods like Gaussian mixtures, and requires only a small change to the typical gradient descent updates with only linear time and memory overhead. We observe that quantization alone is not enough to achieve the desired level of compression, and introduce pruning by adding a standard $\ell_1$ penalty on top of the quantization prior; we demonstrate experimentally that the combined prior yields state-of-the-art compression results on standard benchmark data sets.

## 2 QUANTIZATION BY PARAMETER TYING

We consider the general problem of learning a neural network by minimizing the regularized loss function

$$\mathcal{L}(\boldsymbol{W}) = E_D(\boldsymbol{W}) + \lambda R(\boldsymbol{W}),$$

where $\boldsymbol{W}$ is the set of network parameters of size $N$, $E_D$ is the loss on training data $D$, and $R$ is a function chosen to induce desired properties of learned parameters, e.g. better generalization performance, which cannot be achieved by optimizing $E_D$ alone. $R$ is often chosen to be the $\ell_1$ or $\ell_2$ norm, which encourages sparse parameter vectors or bounded parameter vectors, respectively.

In this work, we achieve quantization with an alternative form of regularization. In a parameter-tied model, $\boldsymbol{W}$ is partitioned into $K$ sets, and parameters in each set are constrained to be equal, i.e., $\boldsymbol{W}$ contains only $K$ distinct values. Formally, for each $k \in \{1, \ldots, K\}$ let $C_k \subseteq \{1, \ldots, N\}$ be disjoint sets, or clusters, of parameter indices, such that $\cup_{k=1}^{K} C_k = \{1, \ldots, N\}$. If the parameters indexed by $C_k$ are required to share the same value, then learning under parameter tying yields the

constrained optimization problem

$$\min_{\boldsymbol{W}} \mathcal{L}(\boldsymbol{W})$$

$$\text{subject to } w_i = w_j, \forall k \in \{1, \ldots, K\}, i, j \in C_k$$

In the neural networks community, parameter tying is a fundamental component of CNNs, where weights are shared across a specific layer. In practice, we often encounter high-dimensional problems with no obvious structure and with no explicit knowledge about how model parameters should be tied. This motivates our goal of discovering parameter tying without prior knowledge, i.e., automatic parameter tying, in which we optimize with respect to both the parameters and the cluster assignments. In general, this problem will be intractable as the number of possible partitions of the parameters into clusters, the Bell number, grows exponentially with the size of the set.

Instead, we consider a relaxed version of the problem, in which parameters are softly constrained to take values close to their average cluster values. To achieve this, we choose the regularizer function $R$ to be a clustering distortion penalty on the parameters, specifically the $k$-means loss $J(\boldsymbol{W}, \boldsymbol{\mu})$, defined to be the sum of the distance between each parameter and its corresponding cluster center,

$$R_{\boldsymbol{\mu}}(\boldsymbol{W}) \triangleq J(\boldsymbol{W}, \boldsymbol{\mu}) \triangleq \frac{1}{2} \sum_n \min_k \|w_n - \mu_k\|_2^2, \tag{1}$$

where $\boldsymbol{\mu} \in \mathbb{R}^K$ is the vector of cluster centers. Note that the $\ell_2$ distance in (1) could be replaced with a variety of other metrics (e.g., if the $\ell_1$ distance is selected, then (1) can be optimized by the $k$-medians algorithm), and $J$ represents a shifted $\ell_2$ norm without the restriction $\boldsymbol{\mu}=0$. From a Bayesian perspective, given a fixed $\boldsymbol{\mu}$, $J(\cdot, \boldsymbol{\mu})$ can be considered as a prior probability over the weights that consists of $K$ independent Gaussian components with different means and shared variances.

While $k$-means has been used for quantization of the weights after training, e.g., see Han et al. (2016) and Gong et al. (2015), we propose to incorporate it directly into the objective as a prior. The hope is that this prior will guide the training towards a *good* parameter tying from which hard-tying (i.e. enforcing the parameter tying constraints) will incur a relatively small loss. Indeed, a primary observation of this paper is that the $k$-means prior (1) proves to be a highly effective one for inducing quantization. The $k$-means prior has fewer parameters/hyperparameters to learn/tune compared to a GMM prior; in addition, it is a more natural prior if we believe that the data is actually generated from a model with finitely many distinct parameters: we expect both priors to perform comparably when the distinct parameters are far apart from each other, but as the clusters move closer together, the GMM prior leads to clusters with significant overlap. In the worst case, the GMM prior converges to a mixture such that each weight has almost exactly the same probability of being generated from each mixture component. This yields poor practical performance. In contrast, $J$ forces each weight to commit to a single cluster, which can result in a lower loss in accuracy when hard-tying. In addition, the maximum likelihood objective for the GMM prior can encounter numerical issues if any of the variances tends to zero, which can happen as components are incentivized to reduce variances by eventually collapsing onto model parameters. This problem can be alleviated by a combination of setting individual learning rates for the GMM and model parameters, annealing the GMM objective (Nowlan & Hinton, 1992), or imposing hyperpriors on the Gaussian parameters to effectively lower-bound the variances (K. Ullrich, 2017). Even with these modifications, significant tuning may still be required to produce good solutions.

## 3 SPARSE AUTOMATIC PARAMETER TYING

Following the approach of Han et al. (2016), if we store the original parameters of a model using $b$-bit floats (typically 16 or 32), and quantize them so that they only take $K$ distinct values, then we only need to store the cluster means, $\boldsymbol{\mu}$, in full precision and the quantized parameters by their index, corresponding roughly to a compression rate of

$$r = \frac{Nb}{N \log_2 K + Kb}. \tag{2}$$

For a parameter-heavy model such that $N \gg K$, the denominator in (2) is dominated by $N \log_2 K$, so most of the savings from quantization come from storing parameter indices with $\log_2 K$ instead of

$b$ bits. Quantization alone, however, is insufficient for state-of-the-art compression. For example, if $b = 32$, a compression rate of well over 100 using the definition above would require $K = 1$ (entire network with a single parameter value) which is infeasible without high accuracy loss. Although compression can be improved by post-processing the quantized parameters with entropy coding, the $k$-means prior does not take advantage of this fact as it encourages equally sized clusters.

We consider another common strategy for compression, network pruning, which results in sparse parameters that can be highly efficiently stored and transmitted using sparse encoding schemes. In our evaluations we use the scheme proposed by (Han et al., 2016) (see Appendix in (K. Ullrich, 2017) for details), in which parameters are first stored in regular CSC or CSR format, whose data structures are further compressed by Huffman coding. Although generally network pruning is orthogonal to quantization, we can achieve both by encouraging a large cluster near zero (referred to as the *zero cluster*): weights in the zero cluster which are effectively zero can be dropped from the model, and nodes that have only zero weights can also be dropped. To this end, we add an additional sparsity-inducing penalty $E_S(\boldsymbol{W})$ to the learning objective resulting in the joint learning objective,

$$\min_{\boldsymbol{W}, \boldsymbol{\mu}} \mathcal{L}(\boldsymbol{W}, \boldsymbol{\mu}) = E_D(\boldsymbol{W}) + \lambda_1 J(\boldsymbol{W}, \boldsymbol{\mu}) + \lambda_2 E_S(\boldsymbol{W}), \tag{3}$$

The case in which $\lambda_2 = 0$, corresponding to no sparsity inducing prior, will be simply referred to as APT (Automatic Parameter Tying) or *plain* APT; the other case as *sparse* APT. In this work, we consider the lasso penalty $E_S(\boldsymbol{W}) \triangleq \|\boldsymbol{W}\|_1$, and find experimentally that this additional penalty significantly increases model sparsity without significant loss in accuracy, for large enough $K$.

We propose a two-stage approach to minimize (3). In stage one, soft-tying, the objective is minimized using standard gradient/coordinate descent methods. In stage two, hard-tying, the soft clustering penalty is replaced with a hard constraint that forces all parameters in each cluster to be equal (as well as parameters in the zero cluster must be zero for sparse APT); the data loss is then minimized using projected gradient descent. Unfortunately, (3) is not a convex optimization problem, even if $E_D$ is convex, as the $K$-means objective $J$ is not convex, so our methods will only converge to local optima in general. We note that unlike the GMM penalty (Nowlan & Hinton, 1992) the $K$-means problem can be solved exactly in polynomial time in the one-dimensional (1-D) case using dynamic programming (Wang & Song, 2011).

In our experiments, we use a fast implementation of 1-D $K$-means that produces a comparable solution to the method proposed in (Wang & Song, 2011), but requires much less time. Also $K$ is selected using a validation set, though nonparametric Bayesian methods could be employed to automatically select $K$ in practice (e.g., DP-means (Kulis & Jordan, 2012)).

## 3.1 Soft-Tying (coordinate descent)

We propose to optimize the (sparse) APT objective $\mathcal{L}$ (3) with a simple block coordinate descent algorithm that alternately optimizes with respect to $\boldsymbol{W}$ and $\boldsymbol{\mu}$. Given $\boldsymbol{\mu}$, optimizing w.r.t. $\boldsymbol{W}$ simply involves gradient descent on $\mathcal{L}$ using backpropagation, with weight decay driving parameters towards their respective cluster centers (as well as $\ell_1$ penalty for sparse APT).

The problem of optimizing w.r.t to $\boldsymbol{\mu}$ given fixed $\boldsymbol{W}$ is solved precisely by the $k$-means algorithm, since $\boldsymbol{\mu}$ only appears in $J$. Typically, the $k$-means problem is equivalently formulated in terms of an arbitrary $\hat{\boldsymbol{\mu}} \in \mathbb{R}^K$ and an $N \times K$ matrix $\mathbf{A}$ of auxiliary variables, such that $a_{nk} \in \{0, 1\}$ indicates whether parameter $w_n$ belongs to cluster $k$, and $\sum_k a_{nk} = 1$; the problem then becomes

$$\min_{\hat{\boldsymbol{\mu}}, \mathbf{A}} J(\boldsymbol{W}, \hat{\boldsymbol{\mu}}, \mathbf{A}) \triangleq \frac{1}{2} \sum_{k=1}^{K} \sum_{n=1}^{N} a_{nk} \|w_n - \hat{\mu}_k\|_2^2,$$

where $\mathbf{A}$ is restricted to satisfy aforementioned constraints. The standard EM-style $k$-means algorithm performs coordinate descent w.r.t $\mathbf{A}$ and $\hat{\boldsymbol{\mu}}$. $\mathbf{A}$ determines a $K$ dimensional linear solution subspace $\mathcal{S}_{\mathbf{A}}$ of the (eventually) hard-tied parameters, and is optimized in the E-step; the M-step updates $\hat{\boldsymbol{\mu}}$ to be the cluster means under assignments $\mathbf{A}$, corresponding to the $K$ unique coordinates of $P_{\mathcal{S}_{\mathbf{A}}}(\boldsymbol{W})$, where $P_{\mathcal{S}_{\mathbf{A}}}(\cdot)$ denotes the orthogonal projection onto $\mathcal{S}_{\mathbf{A}}$. As $k$-means can become expensive for large networks, we found it sufficient to optimize $\hat{\boldsymbol{\mu}}$ only (i.e. updating cluster centers using M-step) after every parameter update, and only run full $k$-means once every 1000 or so parameter updates, and only for long enough (say 100 iterations) for an approximate solution. As shown in our experiments, the frequency of $k$-means updates does not significantly impact the results.

We speed up standard $k$-means by specializing it to 1-D: we take advantage of the fact that comparisons can be done on entire sets of parameters, if we sort them in advance, and operate on partitions of parameter clusters that implicitly define cluster membership. Thus E-step reduces to binary searching between neighboring partitions to redraw cluster boundaries ($O(K \log N)$ time), and M-step updating partition means ($O(N)$ time, but can be greatly reduced by caching partition statistics).

## 3.2 HARD-TYING (PROJECTED DESCENT)

Once the combined objective has been sufficiently optimized, we replace the soft-tying procedure with hard-tying, during which the learned clustering $\mathbf{A}$ is fixed, and parameters are updated subject to tying constraints imposed by $\mathbf{A}$. Prior to hard-tying, the constraints are enforced by the projection $\mathbf{W} := P_{\mathcal{S}_\mathbf{A}}(\mathbf{W})$, i.e. setting parameters to their assigned cluster centers; for sparse APT, we also identify the zero cluster as the one with the smallest magnitude, and enforce sparsity constraint by setting it to zero. $\lambda_1$ and/or $\lambda_2$ can also be annealed; in our experiments we simply fixed them.

In hard-tying, we optimize the data loss $E_D$ via projected gradient descent (the $\ell_1$ loss in soft-tying with sparse APT is dropped in hard-tying): the partial derivatives are first calculated using backpropagation and then all components of the gradient corresponding to parameters in cluster $k$ are set to their average to yield the projected update $\mathbf{W} := \mathbf{W} - \eta P_{\mathcal{S}_\mathbf{A}}(\nabla_\mathbf{W} E_D)$, where $\eta$ denotes the step size. We note that this is distinct from the gradient update suggested by Han et al. (2016), which sums the partial derivatives of all parameters in the same cluster but does not compute the average. This difference arises as Han et al. (2016) only allows weight sharing within each layer, while the projected gradient method we propose can handle weight tying across layers.

Finally, we note that the time overhead of our method per training iteration is essentially linear in $N$, the number of network parameters: the computation of cluster means (hence $P_{\mathcal{S}_\mathbf{A}}(\mathbf{W})$) common to both soft and hard-tying takes linear time, whereas the additional $k$-means steps in soft-tying adds (amortized) constant time per training iteration, as they occur infrequently. Our method's memory requirement is also $O(N)$: the cluster assignments $\mathbf{A}$ are represented as an $N$-vector of integers.

## 4 EXPERIMENTS

In our implementation of soft-tying, we use Tensorflow (Abadi et al., 2015) to optimize $\mathcal{L}$ (3) w.r.t to $\mathbf{W}$. In fact soft-tying can be readily done by auto-differentiation implemented in any modern neural network library. $k$-means is also provided in standard scientific computing libraries, but we implemented the 1D version in C++ for efficiency. We implement hard-tying by first updating $\mathbf{W}$ with $\nabla_\mathbf{W} E_D$ as usual, then performing the assignment $\mathbf{W} := P_{\mathcal{S}_\mathbf{A}}(\mathbf{W})$ (and resetting parameters in the zero cluster to zero for sparse APT).

Unless otherwise mentioned, we initialize the neural network parameters using the method proposed by Glorot & Bengio (2010), and initialize the cluster centers heuristically by evenly distributing them along the range of initialized parameters. As our experiments are concerned with classification problems, we use the standard cross-entropy objective as our data loss. We consider three sets of experiments. First, we use APT on LeNet-300-100 to examine the effect of $k$-means prior, number of clusters versus accuracy, and frequency of $k$-means updates. Inspired by recent work on the generalization performance of neural networks, our second set of experiments aims to understand the effect of APT on the generalization performance of neural networks. Finally, our third set of experiments compares the performance of sparse APT and other state-of-the-art methods under various compression metrics.

## 4.1 ALGORITHMIC BEHAVIOR

We demonstrate the typical behavior of APT using LeNet-300-100 on MNIST. We trained with soft-tying for 20000 iterations, and switched to hard-tying for another 20000 iterations. Figure 1 depicts a typical parameter distribution produced by APT at the end of soft-tying versus training without any regularization, using the same initialization and learning rate. As expected, APT lead to a clear division of the parameters into clusters. Figure 3, in the appendix, illustrates the loss functions and model performance in the experiment, with and without APT. In this demonstration, $K = 8$ appeared sufficient for preserving the solution from soft-tying: switching from soft to hard-tying

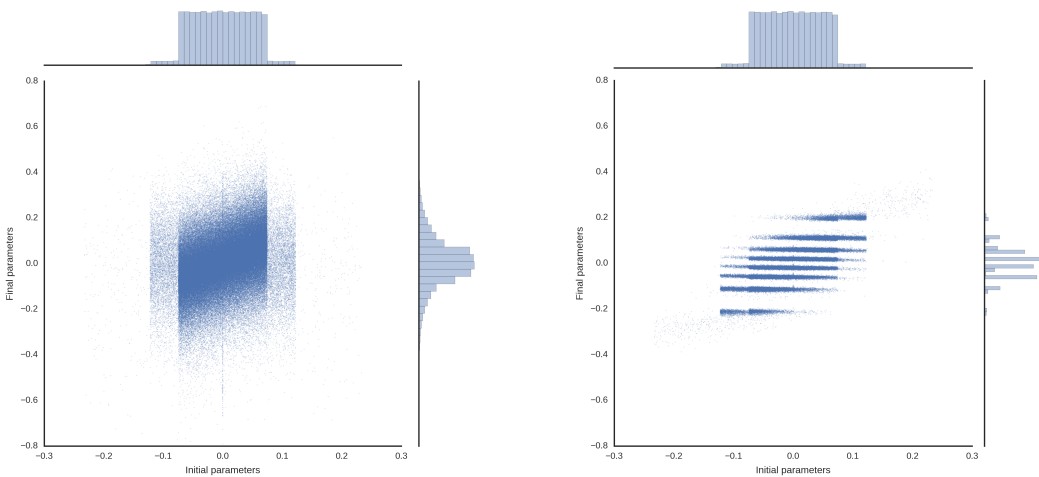

Figure 1: Joint histograms of parameters before and after training, with without (left) and with an additional $k$-means loss (soft-tying APT). The parameters are initialized with scaled uniform distributions proposed in (Glorot & Bengio, 2010) and $K = 8$.

at iteration 20000 resulted in some small loss, and hard-tying was able to gradually recover from it. Generally for a properly chosen $K$, soft-tying does not fundamentally change the convergence speed or final model performance, compared to without APT. However, the loss in accuracy from hard-tying can be significant for small $K$, and decreases with increasing $K$ (the hard-tying phase is generally able to recover from some or all of the accuracy loss for large enough $K$, see 5a). Figure 4 visualizes the cluster trajectory.

We also explored in 5b the effect of coordinate switching frequency on the learning outcome, for which we reran the previous experiments with frequency of $k$-means runs. We observed that APT was generally not sensitive to $k$-means frequency, except for very small $K$, justifying our heuristic for only running $k$-means infrequently. The extreme case of $t = 20000$ corresponds to not running $k$-means, and hence not updating $\mathbf{A}$ at all, effectively randomly (soft) tying the parameters based on their random initial values. Random tying is disastrous for small $K$, which simply cannot effectively cover the range of parameters and induces significant quantization loss. Although specialized training methods, e.g., (Courbariaux et al., 2016), exist for training networks with $K = 2$ or 3, our current formulation of APT cannot effectively quantize with such a small number of clusters.

Finally, we note that the above observations about APT generally hold for sparse APT as well. When $\lambda_1$ and $\lambda_2$ are within a couple of orders of magnitude apart (as in our experiments), the additional $\ell_1$ penalty determines the zero cluster size (hence model sparsity) without strongly impacting the $k$-means loss $J$ or cluster convergence; on the other hand, for a fixed $\lambda_2$, larger $\lambda_1$ (which encourages faster and tighter clustering) would accelerate the growth of the zero cluster and result in higher sparsity and potentially more accuracy loss (which can be alleviated by larger $K$).

### 4.2 EFFECT ON GENERALIZATION

Recently, Zhang et al. (2016) observed that the traditional notion of model complexity associated with parameter norms captures very little of neural networks' generalization capability: traditional regularization methods, like $\ell_2$ (weight decay), do not introduce fundamental phase change in the generalization capability of deep networks, and bigger gains can be achieved by simply changing the model architecture rather than tuning regularization. The paper left open questions of how to *correctly* describe neural network's model complexity, in a way that reflects the model's generalization. In this section, we explore a different notion of model complexity characterized by the number of free parameters in parameter-tied networks, where the tying is discovered through optimization.

To assess whether APT indeed improves model generalization, we compared the performance of APT against a GMM prior, on a toy problem where the latter approach has been shown to be effective. We reproduced the original bit-string shift detection experiment as described in (Nowlan & Hin-

ton, 1992), where the task was to detect whether a binary string was shifted left or right by one bit, with input being the pattern and its shifted version. In the original experiment, all methods (including not using any regularization) were stopped as soon as the training error becomes zero, then test set performance was recorded; we used this same evaluation criteria. We compared the case of no regularization ("early stopping" alone) with $\ell_2$ penalty, APT, and GMM prior. After some initial tuning, we found a common set of SGD step sizes, $\{0.01, 0.03, 0.1, 0.3\}$, and maximum training budget (5000 iterations) such that all methods considered could converge to 0 training error. For $\ell_2$, APT, and GMM, we set the search grid for the regularization parameter $\lambda \in \{10^{-7}, 10^{-6}, \ldots, 10^{-3}\}$, and for GMM in particular, which uses four individual learning rates for the network parameters, mixture means, mixture variances, and mixture proportions, we initialized the hyperparameters as in Nowlan & Hinton (1992) and launched 1280 experiments for all possible combination of parameter step sizes and $\lambda$ for a given $K$. At the end of grid search the best test error for each method was taken across all hyperparameters.

The results are presented in Table 1. As was observed by Zhang et al. (2016), while all methods have essentially zero error on the training data, there appears to be only a mild effect on the test error due to the particular choice of regularization. We performed more evaluations on MNIST and covertype dataset (Blackard & Dean, 1999) with varying network structures: provided that the training process was tuned properly, the different regularization methods (or none at all) resulted in similar final performance for a given network structure; changing the network structure itself, however, had a much stronger impact on performance for all methods. This offers support that automatic parameter tying or norm restriction ($\ell_1$, $\ell_2$) do not act strongly to improve regularization performance.

### 4.3 SPARSITY AND COMPRESSION RESULTS

We compare sparse APT against other neural network compression or pruning methods, including Deep Compression (DC) (Han et al., 2016), Soft Weight Sharing (SWS) (K. Ullrich, 2017), Bayesian Compression (BC) (Louizos et al., 2017), and Sparse Variational Dropout (Sparse VD) (Molchanov et al., 2017) using LeNet-300-100 and LeNet-5-Caffe on MNIST (LeCun et al., 1998), and VGG-16 on CIFAR-10. We use the standard train/test split and form a validation set from 10% of training data, and normalize data by mean/variances of the train set. We perform sparse APT by first soft-tying for a fixed budget of iterations, and then hard-tying for another budget of maximum iterations. In our experiments, we found $K$ in [10, 20] sufficient for networks with a couple of million parameters (or less), and $K$ in [30, 40] sufficient for 10 to 20 million parameters, for achieving $\leq 1\%$ accuracy loss. We tuned $\lambda_1$ and $\lambda_2$ in $[1e-6, 1e-3]$ with grid search and manual tuning.

For compressing LeNets, we trained with Adadelta (Zeiler, 2012) and no data augmentation or other regularization, using soft/hard-tying budgets of 60000/10000 iterations. Unlike in methods such as SWS and BC, we found no loss of accuracy for similar sparsity levels when training from random initialization compared to from a pre-trained network, using largely the same number of iterations. For VGG-16, we used the same amount of data augmentation, dropout, and batch normalization as in (Louizos et al., 2017). We trained with SGD with 0.9 momentum, and decayed the initial learning rate 0.05 by half once the validation accuracy did not improve for 10 consecutive iterations. We observed that training from scratch in this case could not achieve the same accuracy as from a pre-trained solution (about 2% higher error for similar sparsity). We used soft/hard-tying budgets of 80000/20000 iterations, starting with a pre-trained model with 7.3% error.

The results are presented in Table 2. We report the error of the networks on test set, the fraction of non-zero weights, a pruning score based on using CSR format alone (this is defined by equation (9) in Appendix A of (K. Ullrich, 2017)), and maximum compression rate as in (K. Ullrich, 2017). Note that Louizos et al. (2017) evaluate the compression criteria separately for each of their variations of BC, instead of with a single trained network; therefore we report their errors in parentheses following the sparsity/compression statistics as in (Louizos et al., 2017). The maximum compression scores for DC, BC, and Sparse VD were obtained by clustering the final weights into 32 clusters (this achieved the best compression rate (Louizos et al., 2017)). SWS used $K{=}17$ for LeNets, and sparse APT used $K{=}17$ for LeNets and $K = 33$ for VGG-16, corresponding to 16 and 32 distinct non-zero parameter values. When evaluating sparse APT at the same error level as DC on LeNets (1.6% for LeNet300-100 and 0.7% for LeNet-5), we found $K{=}17$ insufficient for achieving such low errors and instead used $K{=}33$ (the same as in DC); the results are shown under "Sparse APT (DC)".

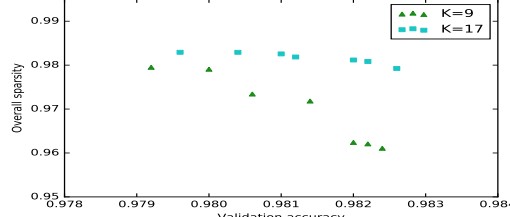

| Regularization | Error | Std. Dev. |
|---|---|---|
| Early Stopping | 84.3 | 3.1 |
| $\ell_1$ | 86.8 | 2.3 |
| $\ell_2$ | 88.3 | 3.7 |
| APT ($K = 5$) | 88.6 | 3.4 |
| APT ($K = 10$) | 88.6 | 3.3 |
| GMM ($K = 5$) | 92.8 | 1.5 |
| GMM ($K = 10$) | 92.4 | 1.7 |

Figure 2: Sparsity versus accuracy trade-off of sparse APT for LeNet-300-100, shown as the Pareto frontier of typical hyper-parameter search results.

Table 1: Average test error of different schemes for the bit-string shift detection problem.

| Network | Method | Error % | $\frac{|w \neq 0|}{|w|}\%$ | Pruning | Max. Compression |
|---|---|---|---|---|---|
| LeNet-300-100 | DC | 1.6 | 8.0 | 6 | 40 |
| | SWS | 1.9 | 4.3 | 12 | 64 |
| | Sparse VD | 1.8 | 2.2 | 21 | 113 |
| | BC-GNJ | - | 10.8 | 9 (1.8) | 58 (1.8) |
| | BC-GNS | - | 10.6 | 9 (1.8) | 59 (2.0) |
| | Sparse APT | 1.9 | 2.1 | 24 | 127 |
| | Sparse APT (DC) | 1.6 | 3.6 | 14 | 77 |
| LeNet-5-Caffe | DC | 0.7 | 8.0 | 6 | 39 |
| | SWS | 1.0 | 0.5 | 100 | 162 |
| | Sparse VD | 1.0 | 0.7 | 63 | 365 |
| | BC-GNJ | - | 0.9 | 108 (1.0) | 572 (1.0) |
| | BC-GNS | - | 0.6 | 156 (1.0) | 771 (1.0) |
| | Sparse APT | 1.0 | 0.5 | 100 | 346 |
| | Sparse APT (DC) | 0.7 | 6.9 | 7 | 45 |
| VGG-16 | BC-GNJ | - | 6.7 | 14 (8.6) | 95 (8.6) |
| | BC-GNS | - | 5.5 | 18 (9.0) | 116 (9.2) |
| | Sparse APT | 8.8 | 5.1 | 10 | 87 |

Table 2: Comparison of sparse APT with other compression and/or sparsity-inducing methods.

Overall, we observe that sparse APT outperforms or performs similarly to all competitors on each data set, with the exception of the BC methods in terms of max compression on LeNet-5 and VGG-16; this occurs even though sparse APT manages to find a sparser solution than both BC variants. The explanation for this is that the maximum compression score uses Huffman coding to compress the cluster indices of quantized parameters in CSR format. As Huffman coding performs best with non-uniform distributions, the primary difference between the sparse APT and the BC solutions is that the BC solutions do not return many equal sized clusters. While our main goal was to achieve sparsity with a small number of parameters, if maximum compression is desired, the variances of the independent Gaussian prior could be tuned to induce a significantly more non-uniform distribution which may yield higher compression rates.

More generally, APT can be used to trade-off between accuracy and sparsity depending on the application. This can be done using a validation set to select the desired performance trade-off. Figure 2 illustrates part of the sparsity/accuracy trade-off curve for two different values of $K$. When $K = 9$, sparsity can be increased at a significant loss to accuracy, while at $K = 17$, additional sparsity can be gained with only moderate accuracy loss. In practice, selecting the smallest value of $K$ that exhibits this property is likely to yield good accuracy and compression. In fact, the existence of such a $K$ provides further evidence that, for a fixed structure, sparsity and quantization has little impact on generalization performance.

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

# A ADDITIONAL EXPERIMENTAL RESULTS

## A.1 APT EXPERIMENTS

Below we illustrate the evolution of cluster centers and change in assignments in the first experiment with LeNet-300-100. Note that the clusters in figure 4a tend to oppose each other, unlike in the case of GMM where they tend to merge; this is a property of $k$-means loss $J$ and independent Gaussian priors. The clusters centers also developed more extreme values during hard-tying.

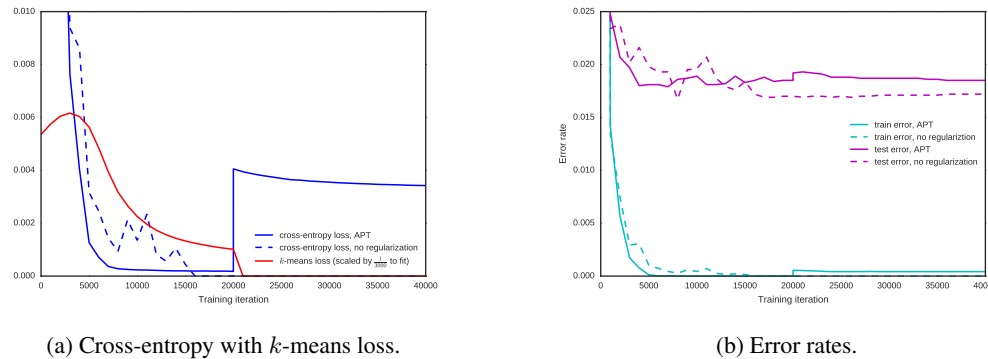

(a) Cross-entropy with $k$-means loss.

(b) Error rates.

Figure 3: Comparison of training with APT (first 20000 iterations soft-tying, last 20000 iterations hard-tying) vs. without regularization on LeNet-300-100, using the same initialization/learning rate.

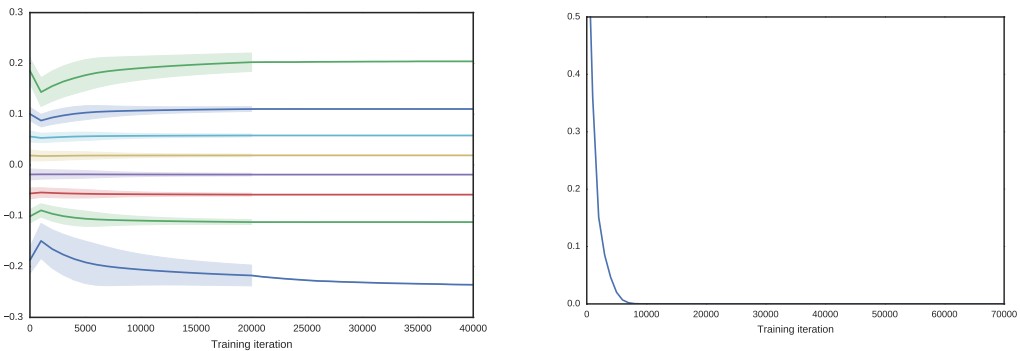

(a) Cluster centers throughout APT training, shaded by 1 standard deviation of points in the cluster.

(b) Change in cluster assignments throughout APT training, as the ratio of parameters that switched assignments in the previous iteration.

Figure 4: Evolution of the clusters in the first APT experiment with LeNet-300-100.

We also examined the effect of $K$ with a series of experiments using LeNet-300-100, in which we learned parameter-tied networks with $K = 2, 4, 8, 16,$ and $32$. We ran soft-tying till convergence for a budget of 30000 iterations, followed by another 20000 iterations of hard-tying. We tuned $\lambda_1$ in the range of $\{$1e-7, 1e-6, ..., 1e-1$\}$, and selected the best model for each $K$ based on validation performance. We did not observe overfitting with either soft-tying or hard-tying, so for simplicity we considered the model performance at the end of their budgeted runs in each phase. Figure 5a displays the best error rates at the end of soft-tying and hard-tying, averaged across 5 random seeds. As can be seen, $K$ did not significantly affect the solution quality from soft-tying; however the accuracy loss involved in switching to hard-tying becomes significant for small enough $K$s, and decreases to zero for $K = 32$.

In another set of APT experiments with similar setup, we examined the impact of $k$-means frequency on model performance for various $K$, in which we vary the number of gradient iterations

$t$ between $k$-means runs, with $t \in \{1, 1000, 5000, 10000, 15000, 20000\}$. Soft/hard-tying were set at 20000/20000 iterations. Here we consider the best end of training (soft-tying followed by hard-tying) error rates after hyperparameter search. As can be seen in 5b, $t$ does not appear to be a sensitive hyperparameter, but model performance does degrade with large $t$, particularly for smaller $K$. Note that $t = 20000$ corresponds to the case of hard-tying (using random cluster assignments) without the soft-tying phase, which generally prevents effective learning except when $K$ is large.

## A.2 SPARSE APT WEIGHT VISUALIZATIONS

Figure 6 visualizes the final weights in LeNet-5's first 20 convolution filters: as can be seen, 11 of them contained zero weights only (thus considered pruned), while the remaining important stroke detectors were quantized. More generally we observed structured sparsity in weights (row/column-wise sparsity for fully connected layers and channel/filter-wise sparsity for conv layers) that result in entire units pruned away, similar to group-sparsity pursued by Wen et al. (2016).

We visualize the first layer weights ($300 \times 784$ matrix) of LeNet-300-100 learned with $\ell_2$, $\ell_1$, and (sparse) APT ($K = 17$, as reported in table 2), all starting from the same random initialization and resulting in similar error rates (between 1.8% and 1.9%).

Figure 7 plots the count of non-zero outgoing connections from each of the 784 input units (shaped as $28 \times 28$ matrix), to the next layer's 300 hidden units. An input unit is considered pruned if all of its outgoing weights are zero; this corresponds to a column of zeros in the weight matrix. Here, sparse APT prunes away 403 of the 784 input units, giving a column-sparsity of 48.6%.

The situation of plain APT is similar to $\ell_2$ and is not shown. In the solutions learned with $\ell_2$ and $\ell_1$, we mark weights with magnitude less than $1e-3$ as zero for illustration, since $\ell_2$ and $\ell_1$ did not result in exactly zero weights.

Figure 8 depicts the first layer weight matrix of LeNet-300-100; each of the 784 input connections to the next layer unit are reshaped as a $28 \times 28$ cell. All colors are on an absolute scale from -0.3 to 0.3 centered at 0; thus a white cell indicates a hidden unit has been disconnected from input and degenerated into a bias for the next layer, corresponding to a sparse row in the weight matrix. Sparse APT results in 76.3% row-sparsity in this case.

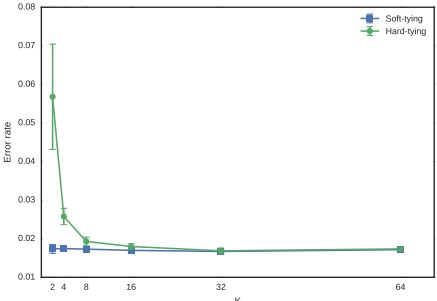

(a) Average error rates (along with 1 standard deviation error bars) at the end of soft-tying, and hard-tying, for LeNet-300-100.

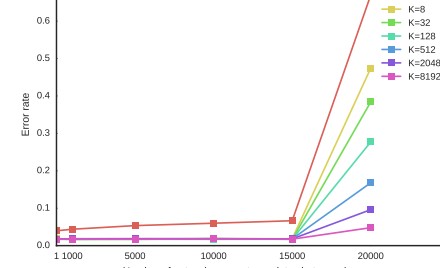

(b) End of training error rates for various $t$ (number of iterations between $k$-means), for various $K$

Figure 5: Effect of varying $K$ and $t$ on learning outcome with APT.

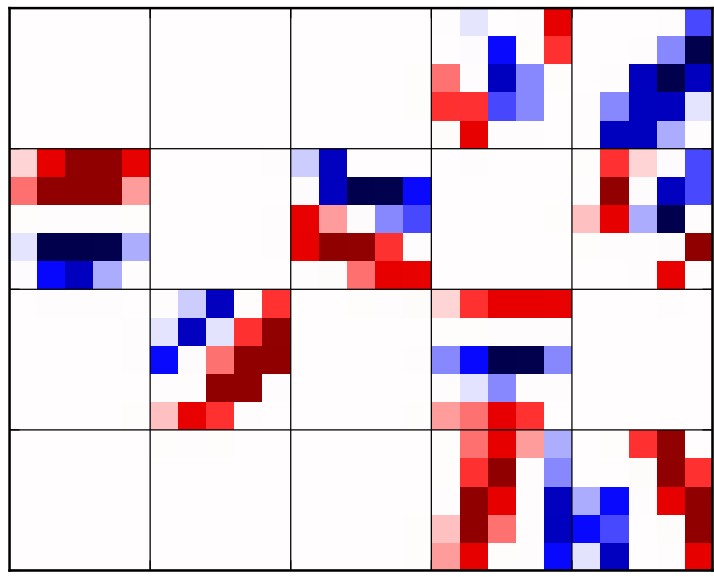

Figure 6: Visualization of the first conv layer in LeNet-5, achieving 1% test error and 99.5% sparsity.

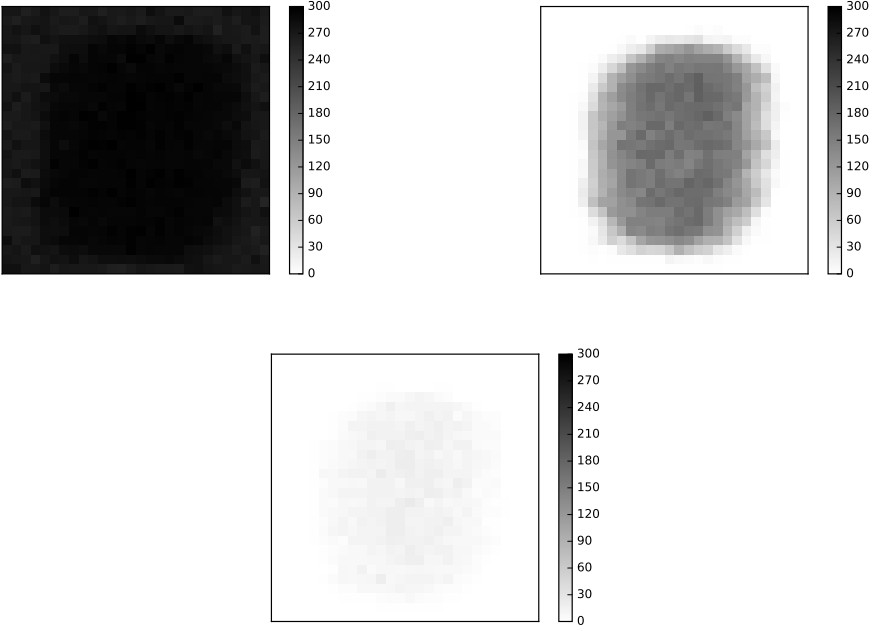

Figure 7: Comparing the the number of input units pruned by $\ell_2$, $\ell_1$, and sparse APT, on LeNet-300-100.

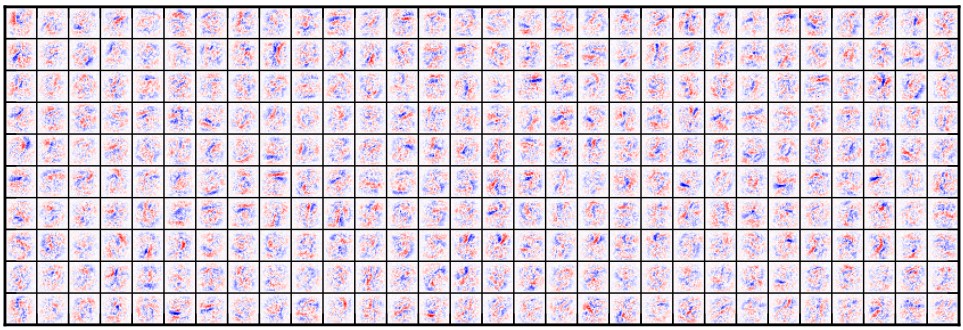

(a) $\ell_2$ regularization.

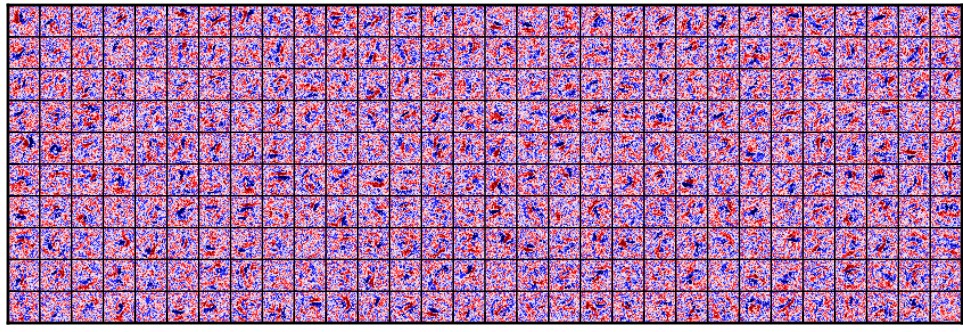

(b) APT regularization, with $K = 8$ quantization levels.

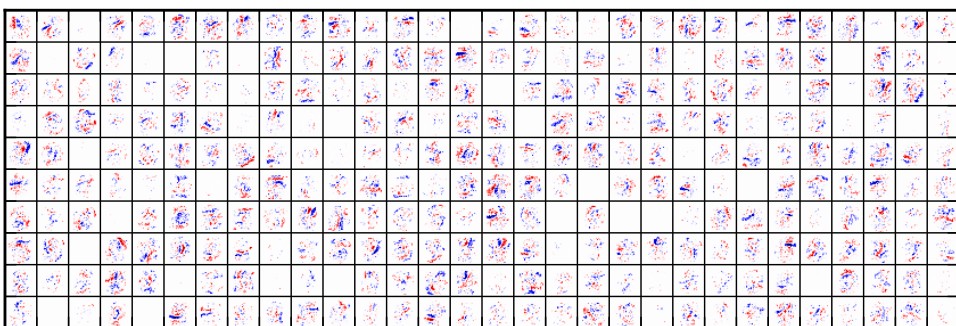

(c) $\ell_1$ regularization.

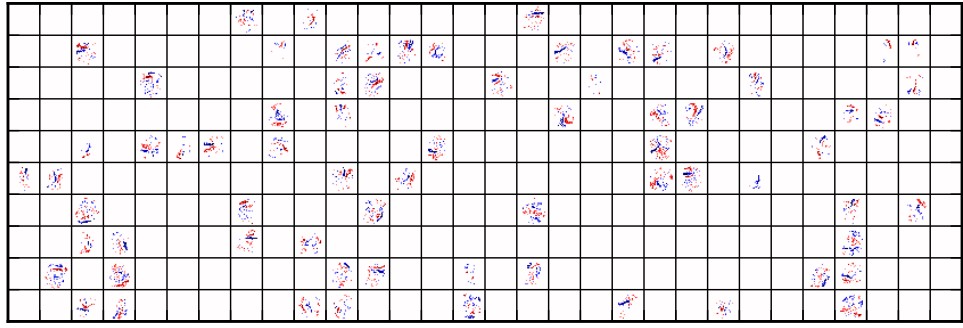

(d) Sparse APT regularization, with 16 non-zero quantization levels.

Figure 8: First layer weight matrix of LeNet-300-100 learned with $\ell_2$, APT, $\ell_1$, and sparse APT.

