# OpenReview forum: "Automatic Parameter Tying in Neural Networks"
_ICLR.cc/2018/Conference — Reject_

### Official Review · AnonReviewer1 · 2017-11-26
**Automatic Parameter Tying in Neural Networks**

**Rating:** 6
**Confidence:** 5

**Review:**

Approach is interesting however my main reservation is with the data set used for experiments and making general (!) conclusions. MNIST, CIFAR-10 are too simple tasks perhaps suitable for debugging but not for a comprehensive validation of quantization/compression techniques. Looking at the results, I see a horrific degradation of 25-43% relative to DC baseline despite being told about only a minimal loss in accuracy. A number of general statements is made based on MNIST data, such as on page 3 when comparing GMM and k-means priors, on page 7 and 8 when claiming that parameter tying and sparsity do not act strongly to improve generalization. In addition, by making a list of all hyper parameters you tuned I am not confident that your claim that this approach requires less tuning.

Additional comments:

(a) you did not mention student-teacher training
(b) reference to previously not introduced K-means prior at the end of section 1
(c) what is that special version of 1-D K-means?
(d) Beginning of section 4.1 is hard to follow as you are referring to some experiments not shown in the paper.
(e) Where is 8th cluster hiding in Figure 1b?
(f) Any comparison to a classic compression technique would be beneficial.
(g) You are referring to a sparsity at the end of page 8 without formally defining it.
(h) Can you label each subfigure in Figure 3 so I do not need to refer to the caption? Can you discuss this diagram in the main text, otherwise what is the point of dumping it in the appendix?
(i) I do not understand Figure 4 without explanation.

---

> ### Author Response · Authors · 2018-01-05
> **Author Rebuttal**
>
> Thanks for the feedback. In table 2 of our revised paper, we added a new experiment that compares with Bayesian compression on VGG16 on CIFAR-10. This is comparable to major existing work (that we’re aware of) on compressing neural networks.  In table 2 we also compare to Deep Compression and GMM prior at the same level of classification accuracy, to address concerns about the accuracy loss in our method.
>
> As with most machine learning methods, some tuning may be needed for optimal performance. In our experiments we simply tried K-1 (number of non-zero parameter values) on a log scale of 4, 8, 16…, and settled on the first K that gave acceptable accuracy loss. The k-means and l1 penalty factors, lamba_1 and lambda_2, were tuned in [1e-6, 1e-3] with a combination of grid search and manual tuning. We believe this is less tuning compared to probabilistic methods like GMM or scaled mixture priors using many more parameters/hyperparameters that are less intuitive and often require careful initialization. In fact, the main reason why we couldn’t compare kmeans with GMM prior ourselves on larger datasets was the latter required significantly more computation and tuning (we were often unable to get it to work).
>
> Regarding additional comments:
> a). Fixed in revised paper.
> b). Fixed in revised paper.
> c). See section 3 of revised paper; as described, at the beginning of the 1D kmeans algorithm, we sort the parameters on the number line and initialize K partitions corresponding to the K clusters; E-step then simplifies to re-drawing the partitions given partition means (requiring K binary-searches), and M-step recalculates partition means (partition sums/sizes are maintained for efficiency).
> d). Fixed in revised paper; we show the typical training dynamics in figures 3 and 4.
> e). Thanks for catching this; we used the wrong image where K=7. See the correct one with K=8 in revised paper.
> f). You mean methods like Optimal Brain Damage and Optimal Brain Surgeon? Admittedly the kmeans distortion is only a rough surrogate to the actual quantization loss, but we found it sufficient for compression; the fact that our method doesn’t use more sophisticated techniques such as second order information means it adds very little overhead to training. Again we’re striving for simplicity and efficiency.
> g). By sparsity we meant the fraction of parameters that are zero.
> h). Fixed in revised paper; added new discussion in results section about the observed structured sparsity (entire units and filters being pruned); we observed this with l1 alone, however, but to a lesser extent and with more accuracy loss.
> i). Fixed in revised paper.

---

### Official Review · AnonReviewer2 · 2017-11-26
**comments on K-means and L1 regularization**

**Rating:** 6
**Confidence:** 4

**Review:**

This is yet another paper on parameter tying and compression of DNNs/CNNs.  The key idea here is a soft parameter tying under the K-means regularization on top of which an L1 regularization is further imposed for promoting sparsity. This strategy seems to help the hard tying in a later stage while keeping decent performance.  The idea is sort of interesting and the reported experimental results appear to be supportive.  However, I have following concerns/comments.

1.  The roles played by K-means and L1 regularization are a little confusing from the paper.  In Eq.3, it appears that the L1 regularization is always used in optimization. However, in Eq.4,  the L1 norm is not included.  So the question is,  in the soft-tying step, is L1 regularization always used?  Or a more general question,  how important is it to regularize the cross-entropy with both K-means and L1?

2. A follow-up question on K-means and L1.   If no L1 regularization, does the K-means soft-tying followed by a hard-tying work as well as using the L1 regularization throughout?

3. It would be helpful to say a few words on the storage of the model parameters.

4. It would be helpful to show if the proposed technique work well on sequential models like LSTMs.

---

> ### Author Response · Authors · 2018-01-05
> **Author Rebuttal**
>
> 1. Please see the revised paper for a clearer discussion of our method. We use kmeans prior alone in our investigation of automatic parameter tying (e.g., in the sections on algorithmic behavior and generalization effect); we always use the L1 norm together with kmeans for the purpose of compression, since quantization alone is not sufficient for state of the art compression (e.g., for 32-bit floats, K=2 roughly gives compression rate of 32; to compress well over 100 times would require K=1, i.e. the entire network using a single parameter value which is infeasible).
>
> 2. The answer is no, as explained above.
>
> 3. As we discussed in section 3 of revised paper, we implemented the sparse encoding scheme proposed by Han, following the detailed appendix in Ullrich. Basically network parameters are stored as CSR matrices (we append the bias from each layer as extra an column to the layer’s weight matrix); the CSR data structures (indices for the position of non-sparse entries, as well as assignment indices) are further compressed with standard Huffman coding.
>
> 4. We haven’t gotten an opportunity to investigate sequential models like LSTMs, but we don’t think anything particular about them may prevent our method from being used. It might require some more tuning to make sure the pull from the cluster centers aren’t strong enough to overpower the gradient signals from data loss, and might require initializing to a pre-trained solution rather than from scratch. That said, we’ve found our method to be rather agnostic towards the nature of different parameters in the network (e.g. weights/biases in all conv/fc layers, along with batch normalization parameters), so it should be able to handle things like memory cells/gates.

---

### Official Review · AnonReviewer3 · 2017-11-27
**Simple and effective compression method, but needs refinement and large-scale experiments**

**Rating:** 6
**Confidence:** 4

**Review:**

As the authors mentioned, weight-sharing and pruning are not new to neural network compression. The proposed method resembles a lot with the deep compression work (Han et. al. 2016), with the distinction of clustering across different layers and a Lasso regularizer to encourage sparsity of the weights. Even though the change seems minimal, the authors has demonstrated the effectiveness on the benchmark.

But the description of the optimization strategy in Section 3 needs some refinement. In the soft-tying stage, why only the regularizer (1) is considered, not the sparsity one? In the hard-tying stage, would the clustering change in each iteration? If not, this has reduced to the constrained problem as in the Hashed Compression work (Chen et. al. 2015) where the regularizer (1) has no effect since the clustering is fixed and all the weights in the same cluster are equal. Even though it is claimed that the proposed method does not require a pre-trained model to initialize, the soft-tying stage seems to take the responsibility to "pre-train" the model.

The experiment section is a weak point. It is much less convincing with no comparison result of compression on large neural networks and large datasets. The only compression result on large neural network (VGG-11) comes with no baseline comparisons. But it already tells something: 1) what is the classification result for reference network without compression? 2) the compression ratio has significantly reduced comparing with those for MNIST. It is hard to say if the compression performance could generalize to large networks.

Also, it would be good to have an ablation test on different parts of the objective function and the two optimization stages to show the importance of each part, especially the removal of the soft-tying stage and the L1 regularizer versus a simple pruning technique after each iteration. This maybe a minor issue, but would be interesting to know: what would the compression performance be if the classification accuracy maintains the same level as that of the deep compression. As discussed in the paper, it is a trade-off between accuracy and compression. The network could be compressed to very small size but with significant accuracy loss.

Some minor issues:
- In Section 1, the authors discussed a bunch of pitfalls of existing compression techniques, such as large number of parameters, local minimum issues and layer-wise approaches. It would be clearer if the authors could explicitly and succinctly discuss which pitfalls are resolved and how by the proposed method towards the end of the Introduction section.
- In Section 4.2, the authors discussed the insensitivity of the proposed method to switching frequency. But there is no quantitative results shown to support the claims.
- What is the threshold for pruning zero weight used in Table 2?
- There are many references and comparisons missing: Soft-to-Hard Vector Quantization for End-to-End Learning Compressible Representations in NIPS 17 for instance. This paper also considers quantization for compression which is related to this work.

---

> ### Author Response · Authors · 2018-01-05
> **Author rebuttal**
>
> Please see the revised paper for a clearer discussion of our proposed method. L1 penalty is indeed used for soft-tying in the sparse-formulation, and yes the hard-tying stage fixes cluster assignments, which is essentially the same as the Hashed Net method except that the assignments are learned from the soft-tying stage, instead of being random.
> Following our discussion in section 3 and 4.1, randomly (hard) tying parameters corresponds to restricting the solution to a random, low dimensional linear subspace; for (especially deep) neural networks that are already hard to train, this extra restriction would significantly hamper learning. The idea is illustrated by figure 5(a) with smaller K and 5(b) for t=20000. Hashed Net effectively uses a very large K with random tying, which poses little/no problem to training, but a larger K would result in degraded compression efficiency for our method. We found soft-tying to be crucial in guiding the parameters to the “right” linear subspace (determined by the assignments, which is itself iteratively improved), such that the projection of parameters onto it is minimized, leading to small accuracy loss when switching to hard-tying; so in this sense we don’t think it’s the same as pre-training the model. That said, starting from a pre-trained solution does seem to make the soft-tying phase easier.
>
> The reference error (no regularization) VGG11 on CIFAR10 in our experiment was about 21%, the same as training with sparse APT from scratch; we apologize for failing to mention that. We replaced this part of experiment with VGG16 (15 million parameters) in the revised paper, to compare with Bayesian compression (Louizos et al. 2017). We agree that the number of parameters (and more generally the architecture) does influence the difficulty of optimization and extent to which a network can be compressed.
>
> Hopefully we made it clear in the revised paper that the kmeans prior for quantization alone is not enough for compression, e.g .K=2 (storing 32-bit floats as 1 bit indices) would roughly give compression rate (without post-processing) of only 32 and likely high accuracy loss with our current formulation. We did a small scale evaluation of l1 penalty alone followed by thresholding for compression, and didn’t find it as effective as kmeans+l1 for achieving sparsity. Note that the Deep Compression work already did an ablation test and reported compression rates with pruning (l1+thresholding) only, and we didn’t find it necessary to repeat this work, since we use the same compression format as theirs.  Please see revised table 2 for our method’s performance at the same classification error as Deep Compression and Soft Weight Sharing (GMM prior), to clear up the concern with accuracy loss in our method.
>
> Regarding the minor issues:
>
> -We feel that many existing methods can be difficult/expensive to apply in practice, and our method has the virtue of being very simple, easy to implement, and efficient (linear time/memory overhead) while achieving good practical performance without much tuning.
>
> -See figure 5(b) added in the appendix.
>
> -As we discuss at the end of sec 3.1 in revised paper, at the end of soft-tying we identify the zero cluster as the one with smallest magnitude, and fix it at zero throughout hard-tying. It is possible to use a threshold to prune multiple clusters of parameters that are near zero, but generally we didn’t find it necessary, as a large zero cluster naturally develops during soft-tying for properly chosen K.
>
> -We weren’t aware of this work; thanks for pointing it out. We’ve added some relevant discussion. The biggest difference compared to our method is that our formulation uses hard assignments even in the soft-tying phase, whereas their method calculates soft-assignment responsibilities of cluster centers for each parameter (similar to the GMM case) and that could take O(NK) time/memory. They achieved smaller accuracy loss on CIFAR-10 than ours, but with K=75 (instead of our 33).  However, it’s not clear how much computation was actually involved.

---

### Author Response · Authors · 2018-01-05
**Updated submission**

We have updated the draft to address the reviewers concerns (plus rearranging to still keep it within the 8 page limit).  Most notably, we have added additional experiments on VGG-16 and improved the clarity of the presentation by adding additional details in both the algorithm description and experimental sections.

---

### Decision · Program_Chairs · 2018-01-29
**ICLR 2018 Conference Acceptance Decision**

**Decision:**

Reject

**Comment:**

This paper presents yet another scheme for weight tying for compressing neural networks, which looks a lot like a less Bayesian version of recent related work, and gets good empirical results on realistic problems.

This paper is well-executed and is a good contribution, but falls below the bar on
1) Discovering something new and surprising, except that this particular method (which is nice and simple and sensible) works well.  That is, it doesn't advance the conversation or open up new directions.
2) Potential impact (although it might be industrially relevant)

Also, the title is a bit overly broad given the amount of similar existing work.